# 2-Butoxytetrahydrofuran and Palmitic Acid from *Holothuria scabra* Enhance *C. elegans* Lifespan and Healthspan via DAF-16/FOXO and SKN-1/NRF2 Signaling Pathways

**DOI:** 10.3390/ph15111374

**Published:** 2022-11-09

**Authors:** Prapaporn Jattujan, Sirin Srisirirung, Warisra Watcharaporn, Kawita Chumphoochai, Pichnaree Kraokaew, Tanatcha Sanguanphun, Prachayaporn Prasertsuksri, Salinthip Thongdechsri, Prasert Sobhon, Krai Meemon

**Affiliations:** 1Chulabhorn International College of Medicine, Rangsit Campus, Thammasat University, Pathum Thani 12120, Thailand; 2Department of Anatomy, Faculty of Medicine, Khon Kaen University, Khon Kaen 40002, Thailand; 3Department of Anatomy, Faculty of Science, Mahidol University, Ratchathewi, Bangkok 10400, Thailand; 4Center for Neuroscience, Faculty of Science, Mahidol University, Ratchathewi, Bangkok 10400, Thailand

**Keywords:** lifespan, oxidative stress, *Holothuria scabra*, *Caenorhabditis elegans*, DAF-16, SKN-1

## Abstract

Extracts from a sea cucumber, *Holothuria scabra*, have been shown to exhibit various pharmacological properties including anti-oxidation, anti-aging, anti-cancer, and anti-neurodegeneration. Furthermore, certain purified compounds from *H. scabra* displayed neuroprotective effects against Parkinson’s and Alzheimer’s diseases. Therefore, in the present study, we further examined the anti-aging activity of purified *H. scabra* compounds in a *Caenorhabditis elegans* model. Five compounds were isolated from ethyl acetate and butanol fractions of the body wall of *H. scabra* and characterized as diterpene glycosides (holothuria A and B), palmitic acid, bis (2-ethylhexyl) phthalate (DEHP), and 2-butoxytetrahydrofuran (2-BTHF). Longevity assays revealed that 2-BTHF and palmitic acid could significantly extend lifespan of wild type *C. elegans*. Moreover, 2-BTHF and palmitic acid were able to enhance resistance to paraquat-induced oxidative stress and thermal stress. By testing the compounds’ effects on longevity pathways, it was shown that 2-BTHF and palmitic acid could not extend lifespans of *daf-16*, *age-1*, *sir-2.1*, *jnk-1*, and *skn-1* mutant worms, indicating that these compounds exerted their actions through these genes in extending the lifespan of *C. elegans*. These compounds induced DAF-16::GFP nuclear translocation and upregulated the expressions of *daf-16*, *hsp-16.2*, *sod-3* mRNA and SOD-3::GFP. Moreover, they also elevated protein and mRNA expressions of GST-4, which is a downstream target of the SKN-1 transcription factor. Taken together, the study demonstrated the anti-aging activities of 2-BTHF and palmitic acid from *H. scabra* were mediated via DAF-16/FOXO insulin/IGF and SKN-1/NRF2 signaling pathways.

## 1. Introduction

Aging is defined as the time-dependent functional decline of an organism, leading to death at the end [1]. Aging is the main risk factor of many neurodegenerative diseases such as Alzheimer’s and Parkinson’s diseases. Due to its irreversible process, there are currently no effective treatments [2]. There are nine remarkable aging hallmarks that play an important role in the aging process including genomic instability, telomere attrition, epigenetic alterations, loss of proteostasis, deregulated nutrient sensing, mitochondrial dysfunction, cellular senescence, stem cell exhaustion, and altered intercellular communication [3]. The root cause behinds these hallmarks is oxidative damage resulting from reactive oxygen species (ROS) derived from both extrinsic and intrinsic sources [3]. Extrinsically, ROS is derived chiefly from environmental pollutants, including smoke, alcohol, and ionizing and UV radiation [4] while, intrinsically, most ROS is derived from the uncoupled electron transport chain in mitochondria [5]. Oxidative stress can be attenuated by natural products from plants and animals, which are invariably used as traditional medicine in Asian countries. In particular, products from sea cucumbers, *Holothuria* spp., have been used as traditional medicines for century [6]. These organisms contain potentially high value-added compounds with therapeutic as well as nutritional properties, such as triterpene glycosides, carotenoids, bioactive peptides, vitamins, minerals, fatty acids, collagens, gelatins, chondroitin sulfates, and amino acids. Scientific studies have reported on the therapeutic efficacies of such compounds from *Holothuria* spp. and suggest their applications for human health. There have been reports on several biological activities of extracts and compounds from *Holothuria* spp., including anti-coagulant, anti-microbial, and anti-oxidant [6,7], anti-aging [8,9], anti-obesity [10], anti-inflammation [11], anti-human glioblastoma cell lines [12], anti-prostate cancer cells [13], anti-breast cancer cells [14], and neuroprotective effects against Parkinson’s disease [15,16] and Alzheimer’s disease [17]. In particular, the crude extracts of *Holothuria scabra* (*H. scabra*) and *H. leucospirota* in ethyl acetate and n-butanol fractions have been shown to extend the lifespan, reduce aging biomarkers, and increase stress resistance in a *Caenorhabditis elegans* (*C. elegans*) model by mediating through *daf-16* (FOXO homolog) activation [8,9]. Our recent study showed that 2-butoxytetrahydrofuran (2-BTHF), a compound isolated from *H. scabra*, could delay paralysis and reduce the level of amyloid-β (Aβ) oligomers in a transgenic *C. elegans* model of Alzheimer’s disease, whereas another compound, palmitic acid, did not delay paralysis in transgenic worms [17]. However, there is no information on whether 2-BTHF, palmitic acid, and other purified compounds from *H. scabra* have anti-aging activity. Therefore, in the present study we aimed to investigate the anti-aging activity of the purified compounds from ethyl acetate and n-butanol fractions of *H. scabra* extracts, using *C. elegans* as a model.

Transgenic and mutants of *C. elegans* have been created to represent human disease genes. Because of its short life span and availability of such mutants, *C. elegans* has been used as a model for the studies related to aging, age-related diseases, mechanisms of longevity, and drug screening in recent decades [18,19]. For aging studies, *C. elegans* possess several signaling pathways amenable to external inputs and epigenetic modifications which are testable for their effects on the delay of the aging process. Specifically, these pathways consist of the insulin/IGF-1 signaling (IIS), AMP-activated protein kinase (AMPK), and mammalian target of rapamycin (mTOR) pathways [18], and a c-Jun N-terminal kinase (JNK), which activates the DAF-16 signaling pathway. Consequently, in this study, we investigated the effects of certain purified compounds from *H. scabra* on lifespan extension, resistance to thermal and oxidative stresses, and the anti-aging related pathways in *C. elegans* models.

## 2. Results

### 2.1. Effects of Purified of H. scabra Extracts on the Lifespan of C. elegans

Five compounds were isolated from ethyl acetate (EA) and n-butanol (BU) fractions of *H. scabra* body walls; these comprised three compounds from ethyl acetate fraction, namely, new diterpene glycosides (holothuria A and B), palmitic acid, and two compounds from butanol fraction, namely, bis (2-ethylhexyl) phthalate (DEHP) and 2-butoxytetrahydrofuran (2-BTHF) [17]. To demonstrate whether these compounds displayed any anti-aging effects, N2 wildtype worms were treated with the compounds at the doses of 1, 5, and 10 μg/mL at 25 °C, and then examined in comparison with the worms treated with a control condition, i.e., 1% DMSO, which was used to solubilize the compounds. The results showed that 5 μg/mL 2-BTHF, 10 μg/mL 2-BTHF, and 1 μg/mL palmitic acid remarkably increased mean lifespans of N2 worms to 12.72 ± 1.90 days, 12.37 ± 2.36, and 11.99 ± 1.86 days, respectively, compared with worms in the control group, whose mean life span was 12.04 ± 1.91, 12.04 ± 1.91, and 11.45 ± 1.87 days, respectively (log-rank, *p* = 0.0012, 0.0192 and 0.0207, respectively). Overall, the lifespans of the treated worms were increased by 5.65%, 2.74%, and 4.72%, respectively, compared to those in the control group (Figure 1, Table 1). By contrast, other compounds did not show any significant extension of the mean lifespans of the treated worms compared to those in the control group.

### 2.2. Effects of Purified H. scabra Compounds on C. elegans’ Resistance to Thermal Stress

The above results showed that *H. scabra* compounds comprising 5 μg/mL 2-BTHF, 10 μg/mL 2-BTHF, and 1 μg/mL palmitic acid could significantly extend the *C. elegans* lifespan. We further analyzed their abilities in promoting resistances to thermal and oxidative stresses. In the thermal stress assay, thirty synchronized L1 larvae were pre-treated with 1% DMSO or 5 μg/mL 2-BTHF or 10 μg/mL 2-BTHF or 1 μg/mL palmitic acid for 72 h at 20 °C. The worms were transferred to S-basal in a 96-well plate with or without compounds and incubated at 35°C for 4 h, and then transferred back to 20 °C for 24 h. The surviving worms were counted daily for 5 days. Pre-treatment with 5 μg/mL 2-BTHF insignificantly increased the thermal stress at day 4 and 5. A 10 μg/mL 2-BTHF slightly increased resistance to thermal stress but with no significant difference at days 3, 4 and 5. Moreover, 1 μg/mL palmitic acid showed an insignificant increase in the survival rates at day 1 and 4 when compared to control (Figure 2). Overall, 5 μg/mL 2-BTHF, 10 μg/mL 2-BTHF, and 1 μg/mL palmitic acid did not alter survival rate of *C. elegans* under thermal stress.

### 2.3. Effects of Purified H. scabra Compounds on C. elegans’ Resistance to Oxidative Stress

In the oxidative stress assay, three sets of experiments, namely, co-treatment, pre-treatment, and all-treatment were performed. In co-treatment, the worms were grown with 1% DMSO for 72 h before being treated with paraquat (PQ) together with 5 μg/mL 2-BTHF or 10 μg/mL 2-BTHF or 1 μg/mL palmitic acid (Figure 3a,d,g). The results showed that 10 μg/mL 2-BTHF could significantly increase the survival rate at 4 h (61.33%) after incubation when compared to the control with 0.1% DMSO (47.31%) (*p* < 0.05) (Figure 3d). Moreover, 1 μg/mL palmitic acid could significantly increase the survival rate at 2 h (96.70%) and 4 h (58.35) after incubation when compared to the control with 0.1% DMSO (2 h (87.92%) and 4 h (47.31%), respectively) (*p* < 0.05) (Figure 3g), whereas 5 μg/mL 2-BTHF could not increase the survival rate (Figure 3a).

In the pre-treatment condition, the worms were pre-treated with 5 μg/mL 2-BTHF or 10 μg/mL 2-BTHF or 1 μg/mL palmitic acid for 72 h and then treated with PQ (Figure 3b,e,h). The results showed that pre-treatment with 5 μg/mL 2-BTHF and 10 μg/mL 2-BTHF and 1 μg/mL palmitic acid could significantly increase the survival rates of the worms after 4 h to 63.91% for 5 μg/mL 2-BTHF, to 74.02% for 10 μg/mL 2-BTHF, and after 2 h to 100% and 4 h to 62.46% for 1 μg/mL palmitic acid after PQ treatments, when compared with the controls (2 h, 87.92%; 4 h, 47.31%), respectively (*p* < 0.05) (Figure 3b,e,h).

In the all-treatment condition, the worms were pre-treated with 5 μg/mL 2-BTHF or 10 μg/mL 2-BTHF or 1 μg/mL palmitic acid for 72 h and then incubated with PQ plus 5 μg/mL 2-BTHF or PQ plus 10 μg/mL 2-BTHF or PQ plus 1 μg/mL palmitic acid (Figure 3c,f,i). The results showed that pre-treatment with 5 μg/mL 2-BTHF and 10 μg/mL 2-BTHF and 1 μg/mL palmitic acid could significantly increase the survival rates at 2, 4,and 6 h for 5 μg/mL 2-BTHF (98.89%, 68.74%, and 45.39%), at 4 h for 10 μg/mL 2-BTHF (65.25%), and at 4 and 6 h for 1 μg/mL palmitic acid (67.72% and 50.36%) after PQ plus each compound when compared with the controls (2 h (87.92%), 4 h (47.31%), and 6 h (32.94%), respectively) (*p* < 0.05) (Figure 3c,f,i).

### 2.4. Effects of Purified H. scabra Compounds on the Pathways That Control Lifespans of C. elegans

The effects of purified compounds on genes encoding insulin/IGF, SIR-2.1, JNK, and SKN-1/NRF2 signaling pathways that control longevity in wildtype and mutant worms were investigated. The N2 wildtype worms were treated with 5 μg/mL 2-BTHF, 10 μg/mL 2-BTHF, and 1 μg/mL palmitic acid in 1% DMSO, while in the control group the N2 wildtype worms were treated only with 1% DMSO. The mean lifespans of N2 treated with 5 μg/mL 2-BTHF, 10 μg/mL 2-BTHF, and 1 μg/mL palmitic acid were 15.20 ± 2.34, 14.50 ± 2.60, and 14.49 ± 2.11 days, respectively, which were significantly increased when compared with the control group (13.79 ± 2.40 days) (*p* < 0.0001, *p* = 0.0174, and *p* = 0.0422, respectively). Thus, lifespans of worms treated with 5 μg/mL 2-BTHF, 10 μg/mL 2-BTHF, and 1 μg/mL palmitic acid, increased by 10.22, 5.15, and 5.08 percent, respectively (Figure 4a, Table 2).

CF1038 (*daf-16*(mu86)) mutant worms were used to investigate whether the compounds could exert their anti-aging effect through the insulin/IGF signaling (IIS) pathway. The worms were treated with 1% DMSO, 5 μg/mL 2-BTHF, 10 μg/mL 2-BTHF, and 1 μg/mL palmitic acid. The mean lifespans of the compound-treated worms were 11.33 ± 1.47, 11.16 ± 1.49, and 11.73 ± 1.48 days, respectively, compared with the DMSO group at 11.42 ± 1.30 days, with no significant difference (*p* = 0.9129, *p* = 0.3820, and *p* = 0.1169, respectively). The percent of increased lifespans were −0.79, −2.28, and 2.71, respectively (Figure 4b, Table 2). Thus, this data demonstrated that the longevity effects of 5 μg/mL 2-BTHF, 10 μg/mL 2-BTHF, and 1 μg/mL palmitic acid require the activity of DAF-16.

VC199 (*sir-2.1*(ok434)) mutant strains were used to test whether anti-aging effects of the compounds were mediated through the SIR-2.1 signaling pathway. VC199 worms treated with 1% DMSO, 5 μg/mL 2-BTHF, 10 μg/mL 2-BTHF, and 1 μg/mL palmitic acid exhibited mean lifespans of 10.51 ± 2.86, 11.08 ± 3.26, 10.13 ± 2.36, and 10.81 ± 2.86 days, respectively. The treatment with 5 μg/mL 2-BTHF appeared to significantly increase the mean lifespan compared to the controls (*p* = 0.0416). For the treatment with 10 μg/mL 2-BTHF and 1 μg/mL palmitic acid, the increases in mean lifespans compared to the control group were not significant (*p* = 0.0799 and *p* = 0.2841, respectively) (Figure 4c, Table 2). Thus, the longevity effect of 10 μg/mL 2-BTHF and 1 μg/mL palmitic acid requires the activity of SIR-2.1. However, 5 μg/mL 2-BTHF may require other signaling molecules for lifespan extension.

To investigate whether the compounds exerted anti-aging effects through the JNK signaling pathway, VC8 (*jnk-1*(gk7)) mutant worms were treated with 1% DMSO, 5 μg/mL 2-BTHF, 10 μg/mL 2-BTHF, and 1 μg/mL palmitic acid. They displayed mean lifespans of 15.51 ± 2.46, 15.32 ± 2.44, 15.91 ± 2.43, and 15.52 ± 2.49 days, respectively, with no significantly difference from the control treated with 1% DMSO (*p* = 0.5789, *p* = 0.1965, and *p* = 0.8594, respectively) (Figure 4d, Table 2). The results indicated that the anti-aging effects of these compounds were also mediated partly through the JNK-1 pathway.

To investigate whether the insulin/IGF signaling (IIS) pathway was affected by the compounds, TJ1052 (*age-1*(hx546)) mutant worms were treated with 1% DMSO, 5 μg/mL 2-BTHF, 10 μg/mL 2-BTHF, and 1 μg/mL palmitic acid. Their mean lifespans were 22.16 ± 3.60, 20.87 ± 3.06, 21.96 ± 2.57, and 22.29 ± 4.39 days, respectively, which were significantly decreased compared to the controls (*p* = 0.0002 and *p* = 0.0136, respectively), while 1 μg/mL palmitic acid did not cause a significant difference in mean lifespans from that of the control group (*p* = 0.6533) (Figure 4e, Table 2). The lifespan percentages were changed by −5.82, −0.90, and 0.59, respectively (Figure 4e, Table 2). These results demonstrated that the longevity effects of 5 μg/mL 2-BTHF, 10 μg/mL 2-BTHF, and 1 μg/mL palmitic acid were mediated partly through the suppression of the IIS pathway with AGE-1 association.

The EU1 (*skn-1*(zu67) IV) mutant strain was also used to test whether SKN-1 signaling pathways play a role as a mediator in anti-aging effects of the compounds. EU1 worms treated with 1% DMSO, 5 μg/mL 2-BTHF, 10 μg/mL 2-BTHF, and 1 μg/mL palmitic acid demonstrated mean lifespans of 12.28 ± 1.76, 12.18 ± 1.97, 12.36 ± 2.10, and 12.29 ± 2.49 days, respectively, which were not significantly different from the control group treated with 1% DMSO (*p* = 0.9737, *p* = 0.2603, and *p* = 0.2695, respectively) (Figure 4f, Table 2). The percentage changes in increased lifespans of the treated groups were −0.81, 0.65, and 0.08, respectively (Figure 4f, Table 2). Thus, the anti-aging effect of these compounds was mediated partly through the SKN-1 pathway.

### 2.5. DAF-16 Translocation

DAF-16 nuclear translocation was demonstrated in the TJ356 strain treated with 1% DMSO, 5 μg/mL 2-BTHF, 10 μg/mL 2-BTHF, and 1 μg/mL palmitic acid. The results showed that 5 μg/mL 2-BTHF strongly induced DAF-16 nuclear translocation (67.78%) with significant difference when compared to intermediate (28.89%) and cytosolic (3.33%) localizations (*p* < 0.01 and *p* < 0.001) (Figure 5b,e). The 10 μg/mL 2-BTHF also showed strong DAF-16 nuclear translocation (65.56%) with significant difference compared to intermediate (26.67%) and cytosolic (7.78%) localizations (*p* < 0.01 and *p* < 0.001) (Figure 5c,e). Furthermore, the results of 1 μg/mL palmitic acid also showed the strong, significant DAF-16 nuclear translocation (63.33%) compared to intermediate (31.11%) and cytosolic (5.56%) localizations (*p* < 0.05 and *p* < 0.001) (Figure 5d,e). In addition, the percentage of DAF-16::GFP nuclear localization in TJ356 worms treated with 5 μg/mL 2-BTHF, 10 μg/mL 2-BTHF, and 1 μg/mL palmitic acid increased significantly when compared to the 1% DMSO group (13.33%) (*p* < 0.0001) (Figure 5e). In contrast, treatment with 1% DMSO showed a significant increase in cytosolic (56.67%) as compared to nuclear translocation (13.33%) (*p* < 0.01) (Figure 5a,e). The positive control was performed by incubating TJ356 worms at 37 °C for 30 min (data not shown).

### 2.6. SOD-3 Expression

To confirm that DAF-16 nuclear translocation could trigger the expression of its downstream genes and superoxide dismutase 3 (SOD-3), an antioxidant mitochondrial enzyme, mutant CF1553 worms expressing the *sod-3p*::GFP reporter were treated with 5 μg/mL 2-BTHF, 10 μg/mL 2-BTHF, and 1 μg/mL palmitic acid. The results showed a significant increase in mean fluorescence intensities of *sod-3p*::GFP (2.17 ± 0.18, 2.35 ± 0.31, and 2.46 ± 0.61, respectively) when compared to the 1% DMSO-treated control group (1.00 ± 0.00) (*p* < 0.05, *p* < 0.01, and *p* < 0.01) (Figure 6a–e).

### 2.7. HSP-16.2 Expression

HSP-16.2, a heat shock protein, is another downstream target of DAF-16. CL2070 worms carrying *hsp-16.2p*::GFP reporter were treated with 5 μg/mL 2-BTHF, 10 μg/mL 2-BTHF, and 1 μg/mL palmitic acid. The results revealed that treatments with 5 μg/mL 2-BTHF, 10 μg/mL 2-BTHF, and 1 μg/mL palmitic acid slightly increased the mean fluorescence intensities of *hsp-16.2p*::GFP (1.39 ± 0.15, 1.12 ± 0.25, and 1.16 ± 0.33, respectively), with no significant difference when compared to the control 1% DMSO group (1.00 ± 0.00) (*p* = 0.1813, *p* > 0.9999, and *p* > 0.9999, respectively) (Figure 7a–e).

### 2.8. GST-4 Expression

To demonstrate that the purified *H. scabra* compounds could upregulate the expression of glutathione S-transferase 4 (GST-4), which is the downstream target of SKN-1, CL2166 worms carrying the *gst-4*p::GFP reporter were treated with 1% DMSO, 5 μg/mL 2-BTHF, 10 μg/mL 2-BTHF, and 1 μg/mL palmitic acid. The results showed that these compounds significantly increased the mean fluorescence intensities of *gst-4*p::GFP (1.19 ± 0.07, 1.19 ± 0.12, and 1.22 ± 0.06, respectively) when compared to the 1%DMSO-treated control group (1.00 ± 0.00) (*p* < 0.05) (Figure 8a–e).

### 2.9. Expression of mRNA Transcripts

To investigate the effects of purified *H. scabra* compounds on the IIS-DAF-16 pathway, the mRNA expression levels of *daf-16*, *hsp-16.2*, and *sod-3* were examined in L1 larvae of N2 worms treated with 1% DMSO, 5 μg/mL 2-BTHF, 10 μg/mL 2-BTHF, and 1 μg/mL palmitic acid for 96 h. The results of qRT-PCR showed that the group treated with 5 μg/mL 2-BTHF (5.80 ± 0.82) exhibited significantly increased *daf-16* mRNA expression level when compared with the 1% DMSO-treated control group (1.00 ± 0.00) (*p* < 0.01) (Figure 9a). In contrast, treatments with 10 μg/mL 2-BTHF (4.20 ± 2.07) and 1 μg/mL palmitic acid (2.75 ± 1.77) exhibited insignificant increase in *daf-16* mRNA expression level when compared with the control (1.00 ± 0.00) (*p* = 0.0750 and *p* = 0.5135) (Figure 9a). In addition, treatment with 5 μg/mL 2-BTHF (3.24 ± 0.67), 10 μg/mL 2-BTHF (3.73 ± 1.68), and 1 μg/mL palmitic acid (4.19 ± 2.29) also insignificantly upregulated mRNA expression levels of *hsp-16.2*, downstream target genes of the IIS-DAF-16 signaling pathway, when compared to the control group (*p* = 0.2887, *p* = 0.1530, and *p* = 0.0842, respectively). Furthermore, treatment with 5 μg/mL 2-BTHF (1.89 ± 0.80), 10 μg/mL 2-BTHF (1.80 ± 1.26), and 1 μg/mL palmitic acid (2.60 ± 1.79) insignificantly enhanced mRNA expression levels of *sod-3*, downstream target genes of the IIS-DAF-16 signaling pathway, when compared to the control group (*p* > 0.9999, *p* > 0.9999, and *p* = 0.3902, respectively). Moreover, we examined the mRNA expression of *gst-4*, a downstream target gene of the SKN-1 signaling pathway. The result showed that treatments with 5 μg/mL 2-BTHF (2.19 ±0.68), 10 μg/mL 2-BTHF (1.98 ± 0.95), and 1 μg/mL palmitic acid (2.17 ± 0.96) slightly, but not significantly, increased the expression level of *gst-4* when compared with 1% DMSO-treated control (*p* = 0.2708, *p* = 0.4505, and *p* = 0.2823, respectively) (Figure 9b–d).

## 3. Discussion

Our present study demonstrated that BUP2 and EAP3 compounds purified from butanol and ethyl acetate fractions of the body wall of *H. scabra* significantly increased the mean lifespan of wildtype *C. elegans*, as well as protected them against oxidative stress but not thermal stress. Similarly, crude extracts from the sea cucumbers *H. scabra* [8] and *H. leucospilota* [9] also showed protective capacity against oxidative stress in a *C. elegans* model. BUP2 and EAP3 compounds were determined in our previous study to be 2-butoxytetrahydrofuran (2-BTHF) and palmitic acid, respectively [17]. Although the lifespan extension of the compound-treated worms found in our study was significant when compared to the control worms, the lifespans of the control as well as the treated worms were not as high as those reported in other studies [20,21]. This may be due to the temperature effect as it was shown that lifespan was greater when the worms were cultured at 22 °C than at 25 °C. Furthermore, the higher concentration of DMSO (1%) used in our study to completely dissolve the compounds might affect the worms’ lifespan as compared to other studies which used a lower concentration of DMSO to dissolve their compounds. However, the mean lifespan from our study is also consistent with the studies by Rangsinth et al., 2019 and Guha et al., 2012, which reported the mean lifespan of N2 were 12.41 ± 0.12 [22] and 11.83 ± 0.946 [23] days, respectively.

Since there are several signaling pathways, including JNK, DAF-16/FOXO, and SKN-1/NRF2 pathways, associated with lifespan extension (20), we further demonstrated that 2-BTHF and palmitic acid extended lifespan of *C. elegans* through these pathways by using mutant worms whose genes encoding for insulin/IGF-1 (IIS), SIR-2.1, JNK, and SKN-1 pathways were deleted [24,25,26,27,28]. The results revealed that treatments with 2-BTHF and palmitic acid could not extend the lifespans of *daf-16* and *age-1* mutant worms, suggesting that the longevity effects of 2-BTHF and palmitic acid were mediated through the activity of the insulin/IGF signaling pathway. Similarly, 2-BTHF and palmitic acid could not extend the lifespans of the mutant worms *sir-2.1*, *jnk-1*, and *skn-1*, which also indicates that SIR-2.1, JNK and SKN-1 signaling pathways are necessary for these compounds to extend lifespan as shown for certain compounds in other studies [21,29].

We further investigated the effects of the compounds on the insulin/IGF signaling pathway by qRT-PCR as well as expressions of DAF-16 and its downstream targets including *hsp-16.2* and *sod-3* genes by GFP labelling [30]. It was found that treatment with 2-BTHF and palmitic acid significantly induced DAF-16::GFP nuclear translocation and enhanced SOD-3::GFP, but insignificantly increased HSP-16.2::GFP expression levels, suggesting that the compounds activate the DAF-16 insulin/IGF signaling pathway for lifespan extension and oxidative stress resistance. Meanwhile, these compounds also significantly enhanced GST-4::GFP expression level in the transgenic CL2166 strain, which is a downstream target of the SKN-1 signaling pathway [31]. In support of the latter finding, 2-BTHF and palmitic acid did not extend lifespan of the EU1 (*skn-1*(zu67) IV) mutant strain, suggesting that 2-BTHF and palmitic acid also require SKN-1 activity to prolong lifespan. Taken together, the SKN-1 signaling pathway, through the induction of GST-4 expression, is necessary for mediating the lifespan extension activities of 2-BTHF and palmitic acid.

Cytoprotective effects of ether compounds have been reported, including antifungal, anti-inflammation, and anticancer [32]. Another study also reported that anandamide and 2-arachidonylglycerol ether modulated an anti-oxidative defense mechanism by activating expressions of catalase, superoxide dismutase, and glutathione peroxidase [33]. In addition, anandamide and 2-arachidonylglycerol ether exhibited neuroprotective effects against Aβ-induced neurotoxicity through MAPK activation [34]. A recent study revealed that 2-BTHF, a small cyclic ether, reduced amyloid-β (Aβ) aggregation and attenuated Aβ proteotoxicity in a *C. elegans* model of Alzheimer’s disease [17]. Heat shock factor 1 (HSF-1) was identified as a possible target of 2-BTHF that could contribute to enhanced expression of autophagy-related genes essential for the breakdown of the Aβ aggregate, hence reducing its toxicity. Therefore, it was proposed that 2-BTHF from *H. scabra* protects *C. elegans* from Aβ toxicity by reducing its aggregation via an HSF-1-regulated autophagic route, and the compound was suggested as a potential candidate for Alzheimer therapy [17]. In addition, bromophenol bis (2,3,6-tribromo-4,5-dihydroxybenzyl) ether (BTDE), a small ether isolated from marine algae, has been reported to mediate a similar effect as 2-BTHF by increasing activity of SOD to protect against oxidative damage [35]. Considering the structure–activity relationship, a cyclic ether is generally composed of a cyclic ring containing non-carbon atom such as nitrogen, oxygen, and sulfur in its ring structure [36]. This enables various heterocyclic compounds to be potent antioxidants that prevent oxidative damage, since they can prevent free radicals from taking electrons from other biomolecules. In addition to activating the IIS-DAF-16 and SKN-1 signaling pathways, it is possible that 2-BTHF directly provides an electron to free radicals and protects *C. elegans* against oxidative damage, an important hallmark of the aging process.

It has been shown that sea cucumbers have high saturated fatty acids as well as ω-3/ω-6 fatty acids that enhance human health. Palmitic acid was found to be among the most dominant saturated fatty acid in sea cucumbers as its level ranged from 9.60 to 28.30% in fresh form and decreased to 3.44–17.41% in processed form [37]. Palmitic acid (16:0) is also the most common saturated fatty acid in the human body, which can be obtained from food or generated in the body from other fatty acids, carbohydrates, and amino acids. It is also a major component of palm oil as well as meat, dairy products, cocoa butter, and breast milk. In membrane phospholipids and adipose triacylglycerols, palmitic acid accounts for 20–30% of total fatty acids. Disruption of palmitic acid homeostatic balance has been linked to a variety of physio-pathological disorders, including atherosclerosis, neurological illnesses, and cancer. These diseases are frequently linked to unregulated palmitic acid endogenous production, regardless of dietary contribution. A previous study revealed that palmitic acid can stimulate inflammation and insulin resistance in a human SH-SY5Y neuroblastoma cell line [38]. Moreover, palmitic acid can also induce insulin resistance of macrophages in vitro by upregulating galectin-3 (*Gal-3*) expression and promoting the TLR4/phosphorylated-NF-B signaling pathway [39].

However, the effects of a high-fat diet, primarily palmitic acid, remain controversial in development of cardiovascular diseases (CVD), obesity-related disorders, and cancer [40]. Although the negative effects of palmitic acid on lifespan have been shown in Drosophila and rodents [41,42], the present study revealed that palmitic acid increased lifespan in a *C. elegans* model via DAF-16/IIS and SKN-1 signaling pathways. The lifespan extension ability of palmitic acid may be associated with its antioxidant properties as demonstrated in our study and other studies [43,44]. Another point of concern could be the optimal ratio between levels of saturated and unsaturated fatty acids in individual organisms that also provide beneficial effects on health [45]. Dose-dependent effects known as hormesis could also be another factor involved. The beneficial effects of the compounds could be found when applied at low dose, whereas at high dose it presents negative adverse effects. Therefore, to obtain the optimal beneficial effects of dietary palmitic acid, those factors should be taken into consideration, while more precise studies should also be conducted before findings are applied for future usage.

## 4. Materials and Methods

### 4.1. H. scabra Extraction and Purification

The sea cucumber *H. scabra* was obtained from the Coastal Fisheries Research and Development Center, Prachuap Khiri Khan, Thailand. The body walls of *H. scabra* were collected and underwent the extraction and purification processes to obtain pure compounds from ethyl acetate (EA) and n-butanol (BU) fractions [16,17]. These procedures were ethically conducted under the Mahidol University-Institute Animal Care and Use Committee (MU-IACUC; MUSC60-049-399).

For extraction and purification processes, freeze-dried *H. scabra* samples were extracted with 95% EtOH to obtain EtOH extract. Then, EtOH extract was eventually partitioned using solvent partition by *n*-hexane, ethyl acetate, and butanol, affording an *n*-hexane fraction, an ethyl acetate fraction, and a butanol fraction, respectively. The ethyl acetate fraction (EA) was subsequently fractionated by column chromatography (CC) using Sephadex LH-20 with a 100% MeOH condition. Then the elute was examined by Thin Layer Chromatography (TLC) to obtain 3 fractions (EA1, EA3, and EA3). Fraction EA1 was subjected to CC *n*-hexane-EtOAc (80:20) to afford purified compound 3 (EAP3). Fraction EA2 was subjected to CC Reversed-phase MeOH-H_2_O (80:20) with isocratic condition to afford purified compound 1 (EAP1) and compound 2 (EAP2). The butanol fraction was fractionated by CC using a gradient solvent system of *n*-hexane-EtOAc (80:20), (70:20), (50:50), and EtOAc fraction with increasing amounts of more polar solvents of the eluates examined by TLC, to obtain 4 combined fractions (BU1, BU2, BU3, and BU4). Fraction BU2 was sequentially subjected to CC on silica gel by using CH_2_Cl_2_-MeOH (100:1) to yield subfractions 1–3. Subfraction 3 was further chromatographed using an n-hexane-EtOAc (80:20) condition, obtaining purified BUP1 compound. Subfraction 2 was re-chromatographed using an n-hexane-EtOAc (80:20) condition to yield purified BUP2 compound.

Then, the chemical structures of all 5 purified compounds were analyzed using ^13^C/^1^H-NMR, DEPT 135, electrospray ionization time-of-flight mass spectra (ESI-TOF-MS), and HMBC and COSY spectra as reported in our previous publications [16,17]. NMR spectra were recorded on a Bruker AVANCE 400 FT-NMR spectrometer (Zurish, Switzerland), operating at 400 MHz for ^1^H-NMR and 100 MHz for ^13^C-NMR. A tetramethyl-silane (TMS) was used as an internal standard. For mass spectroscopy, ESI-TOF-MS were measured with a Bruker micrOTOF-QII mass spectrometer (Billerica, MA, USA). CC was carried out using either Merck silica gel 60 (particle size less than 0.063 mm), reversed-phase RP C-18 (40–63 µm), or Pharmacia Sephadex LH-20. For TLC, Merck precoated silica gel 60 F254 plates and RP-TLC, RP-18 F254 precoated on an aluminum plate (E. Merck) were applied. Spots on a TLC plate were detected under UV light and sprayed with anisaldehyde-H_2_SO_4_ reagent followed by heating.

Finally, the compounds EAP1, EAP2, EAP3, BUP1, and BUP2 were identified and characterized as diterpene glycosides (holothuria A and B), palmitic acid, bis (2-ethylhexyl) phthalate (DEHP), and 2-butoxytetrahydrofuran (2-BTHF), respectively (17). The purity percentage of each compound was assessed by qNMR to be more than 90%. The compounds were finally dissolved with DMSO to obtain the final concentration at 1% for improved solubility of the compounds.

### 4.2. C. elegans Strains and Growth Conditions

*C. elegans* strains were composed of Bristol N2 (wild-type), CF1038 (*daf-16*(*mu86*)), VC199 (*sir-2.1*(*ok434*)), VC8 (*jnk-1*(*gk7*)), TJ1052 (*age-1*(*hx546*)), TJ356 (*daf-16::gfp* (zIs356 (pDAF-16::DAF-16-GFP;rol-6)), CF1553 (muIs84 [(pAD76) *sod-3*p::GFP+rol-6(su1006)]), CL2070 (dvIs70 [*hsp-16.2*p::GFP + rol-6(su1006)]), and CL2166 (dvIs19 [(pAF15)*gst-4*p::GFP::NLS] III). The aforementioned strains were obtained from the *Caenorhabditis* Genetics Center, University of Minnesota, USA. All strains were maintained on nematode growth medium (NGM) plates seeded with *E. coli* OP50 at 20 °C for an entire experiment. The *C. elegans* protocol was ethically performed under the guidelines of MU-IACUC (MUSC60-047-397) and Thammasat University-Institute Animal Care and Use Committee (Protocol Number: 020/2562).

### 4.3. Lifespan Assays

Synchronized N2 L1 larvae were incubated at 20 °C in the NGM plate overlaid with OP50 for approximately 37 h to become L4 larvae. Subsequently, synchronized L4 larvae were transferred by platinum wires to NGM plates containing 150 mM 5-fluoro-2′-deoxyuridine (FUDR; Sigma Aldrich, St. Louis, MO, USA), which was conducive to inhibit larvae’s progeny, followed by feeding with OP50 that were mixed with 1% DMSO, along with 1, 5, and 10 μg/mL of several purified *H. scabra* compounds (diluted in 1% DMSO). The plates were finally kept in an incubator at 25 °C [23]. Approximately, a total number of 120 worms were analyzed (3 plates per group and 40 worms per plate). Live, dead, and censored worms were counted daily from the first day until all worms died. The number of live and dead worms was calculated for statistics represented by means ± SD.

### 4.4. Stress Resistance Assays

It has been shown that increased reactive oxygen species (ROS) could affect longevity. From lifespan assay results, the purified *H. scabra* compounds that significantly increased the longevity of wild-type worms were 5 μg/mL 2-BTHF, 10 μg/mL 2-BTHF, and 1 μg/mL palmitic acid. There are also several stresses that can cause overproduction of ROS. To test the effect of compounds on lowering ROS, therefore, we selected thermal stress and oxidative stress for investigation.

In the thermal stress assay, synchronized N2 L1 larvae were seeded onto NGM plates with FUDR that were overlaid by OP50 mixed with 1% DMSO or 5 μg/mL 2-BTHF or 10 μg/mL 2-BTHF or 1 μg/mL palmitic acid. The worm plates were kept at 20 °C for 72 h; then, 30 synchronized adult worms were transferred by platinum wires to an S-basal medium containing *E. coli* OP50 and *H. scabra* compounds in the same concentration in 96-well plates (10 worms/well and total of 3 wells for each group). The 96-well plates were incubated at 35 °C for 4 h and then transferred back to 20 °C. After 24 h, the live and dead worms were counted daily for 5 days.

In the oxidative stress assay, paraquat (PQ), a chemical that can generate oxidative stress in organisms, was used in this experiment. Synchronized L1 larvae of N2 were incubated on NGM/FUDR plates containing OP50 mixed with only 1% DMSO alone or the *H. scabra* compounds (5 μg/mL 2-BTHF or 10 μg/mL 2-BTHF or 1 μg/mL palmitic acid) at 20 °C for 72 h. Then, 30 synchronized adult worms were transferred by platinum wires to an S-basal medium containing *E. coli* OP50 and 250 mM paraquat in a 96-well plate under the following conditions [8]; the first group was worms pre-treated with 0.1% DMSO and then transferred to the wells containing 0.1% DMSO, or 250 mM paraquat [20], or 5 μg/mL 2-BTHF plus 250 mM paraquat, or 10 μg/mL 2-BTHF plus 250 mM paraquat, or 1 μg/mL palmitic acid plus 250 mM paraquat. The second group was worms pre-treated with 5 μg/mL 2-BTHF and then treated in the wells containing 5 μg/mL 2-BTHF or 250 mM paraquat or 5 μg/mL 2-BTHF plus 250 mM paraquat. The third group was worms pre-treated with 10 μg/mL 2-BTHF and then treated in the wells containing 10 μg/mL 2-BTHF or 250 mM paraquat or 10 μg/mL 2-BTHF plus 250 mM paraquat. The fourth group was worms pre-treated with 1 μg/mL palmitic acid and subsequently treated in the wells containing 1 μg/mL palmitic acid or 250 mM paraquat or 1 μg/mL palmitic acid plus 250 mM paraquat. The live or dead worms were counted every 2 h for 10 h. The experiments were performed in triplicate.

### 4.5. Lifespan Assays of Mutant Worms

Various *C. elegans* strains were examined for anti-aging effect by treatments with 5 μg/mL 2-BTHF, 10 μg/mL 2-BTHF, and 1 μg/mL palmitic acid, and with 1% DMSO for the control worms. *C. elegans* strains analyzed in the experiment consisted of N2 wild-type and several mutant strains, i.e., CF1038, *daf-16*(mu86); VC199, *sir-2.1*(ok434); VC8, *jnk-1*(gk7); TJ1052, *age-1*(hx546); and EU1, *skn-1*(zu67). Synchronized L1 larvae of each strain were transferred to NGM plates in the presence of OP50 and were later incubated at 20 °C for approximately 37 h to become L4 larvae. Next, forty synchronized L4 larvae from each strain were transferred to NGM/FUDR plates seeded with OP50 mixed with 1% DMSO or 5 μg/mL 2-BTHF or 10 μg/mL 2-BTHF or 1 μg/mL palmitic acid and incubated at 22 °C [46]. The lifespan assay was performed at 22 °C since some mutant worms had shortened lifespan when incubated at high temperature, i.e., 25 °C. Each group of worms was performed in three plates with a total number of about one-hundred and twenty worms. The live, dead, and censored worms were recorded and counted daily until all worms died. The live worms were analyzed for survival rate of worms by using the log-rank (Mantel-Cox) test of the GraphPad Prism software.

### 4.6. DAF-16 Translocation and SOD-3, HSP-16.2, and GST-4 Expression in Transgenic Strains of C. elegans

To examine nuclear translocation of *daf-16* and its downstream gene, TJ356 strain *daf-16*::GFP (zIs356 (pDAF-16::DAF-16-GFP;rol-6)) was analyzed. To explore the expression of SOD-3 and HSP-16.2, CF1553 strain muIs84 [(pAD76) *sod-3*p::GFP+rol-6(su1006)], and CL2070 strain dvIs70 [*hsp-16.2*p::GFP + rol-6(su1006)], which are downstream genes of DAF-16, were used for study, respectively. CL2166-dvIs19 [(pAF15) *gst-4*p::GFP::NLS] III was used to examine the expression of GST-4 as a downstream gene of SKN-1.

After the synchronization of *C. elegans*, L1 larvae of TJ356, CF1553, CL2070, and CL2166 strains were transferred to NGM/FUDR plates containing OP50 mixed with 1% DMSO or 5 μg/mL 2-BTHF or 10 μg/mL 2-BTHF or 1 μg/mL palmitic acid and incubated at 20 °C. Worms were collected after incubating for 96 h. To initiate DAF-16 nuclear translocation, TJ356 worms were incubated at 37 °C for 30 min as part of the positive control. Each group had thirty worms randomly put onto microscope slides, which were subsequently coated with 2% agarose, anesthetized with 2% sodium azide, and covered with coverslips. A fluorescence microscope was used to capture the GFP intensity of DAF-16, SOD-3, HSP-16.2, and GST-4 expression in the worms (Olympus BX53). DAF-16::GFP nuclear translocation was classified as nuclear, cytosolic, or intermediate, and the results were compared to the control [24,47]. The GFP intensities of SOD-3, HSP-16.2, and GST-4 expression of *C. elegans* were measured at head region (for SOD-3 and HSP16.2) and whole body (for GST-4) using the ImageJ program. The experiments were performed in triplicate.

### 4.7. Gene Expression Analysis by Quantitative RT-PCR

Quantitative RT-PCR was used for analyzing the expression of *daf-16* and its down-stream genes, *hsp-16.2*, *sod-3*, and *gst-4*, in wild-type N2. The NGM/FUDR plates containing OP50 mixed with 1% DMSO or 5 μg/mL 2-BTHF or 10 μg/mL 2-BTHF or 1 μg/mL palmitic acid were prepared prior to being transferred to synchronized N2 L1. Then, the NGM/FUDR plates underwent incubation at 20 °C for 96 h. Approximately 300 worms from each group were transferred from NGM/FUDR plates to an eppendorf tube. Consecutively, M9 buffer was applied to wash the synchronized worms for 3 times in order to remove bacteria, and supernatants were then removed. In the next step, the tubes containing worms were immediately immersed into the liquid nitrogen. RNAs of worms were extracted by using RNA miniprep kit with column (Qiagen, Germany), measured by nanodrop and kept at −80 °C. iScript^TM^ Reverse Transcription Supermix R (Cat. #170-8841, Bio-Rad, Hercules, CA, USA) was used for converting RNA into cDNA. Then, Quantitative PCR (qPCR) was performed by using SsoFast™ EvaGreen^®^ Supermix with Low ROX (Cat. #172-5211, Bio-Rad, Hercules, CA, USA), specific qPCR primers, and the converted cDNA. The specific qPCR primers for *daf-16*, *hsp-16.2*, *sod-3*, *gst-4*, and *act-1* (an internal control) are shown in Table 3. The enzyme activation required 1 cycle together with 95 °C for a period of 30 s of qPCR condition. This was followed by the denaturation process which required 44 cycles at 95 °C for 5 s, and 60 °C for 30 s for annealing. Ultimately, the reaction was discontinued with the condition of 75 °C for 30 s. The aforementioned qPCR steps were performed again for three times using independent RNA preparation. The ΔΔCt method was used to quantify the data.

### 4.8. Data Analysis

The statistics were analyzed by GraphPad Prism software (GraphPad Software, Inc., La Jolla, CA, USA). The survival curves were calculated for *p* value by using the log-rank (Mantel-Cox) test. All results were expressed as means ± standard deviation (SD). The results from thermal and oxidative stresses and nuclear translocation were analyzed for significant differences by two-way ANOVA. The significant differences were analyzed by one-way ANOVA for GFP expressions and gene expressions. The value of *p* < 0.05 indicated statistical significance.

## 5. Conclusions

In conclusion, BUP2 and EAP3 compounds of *H. scabra* identified as 2-butoxytetrahydrofuran (2-BTHF) and palmitic acid could extend lifespan and increase stress resistance in the *C. elegans* model. Treatment with 2-BTHF and palmitic acid enhanced DAF-16 nuclear translocation, and upregulated the expressions of *daf-16*, *hsp-16.2*, and *sod-3* transcripts and SOD-3::GFP expression, suggesting lifespan extension mediated via the DAF-16/FOXO signaling pathway. In addition, 2-BTHF and palmitic acid could enhance *gst-4* transcript and GST-4::GFP expression, and abolish lifespan extension of the *skn-1* mutant strain, indicating that they might also promote longevity through the SKN-1 signaling pathway. Therefore, the present data demonstrated the anti-aging potential of 2-BTHF and palmitic acid, which could be applied for health promotion to delay aging and reduce the risk of age-associated diseases.

## Figures and Tables

**Figure 1 pharmaceuticals-15-01374-f001:**
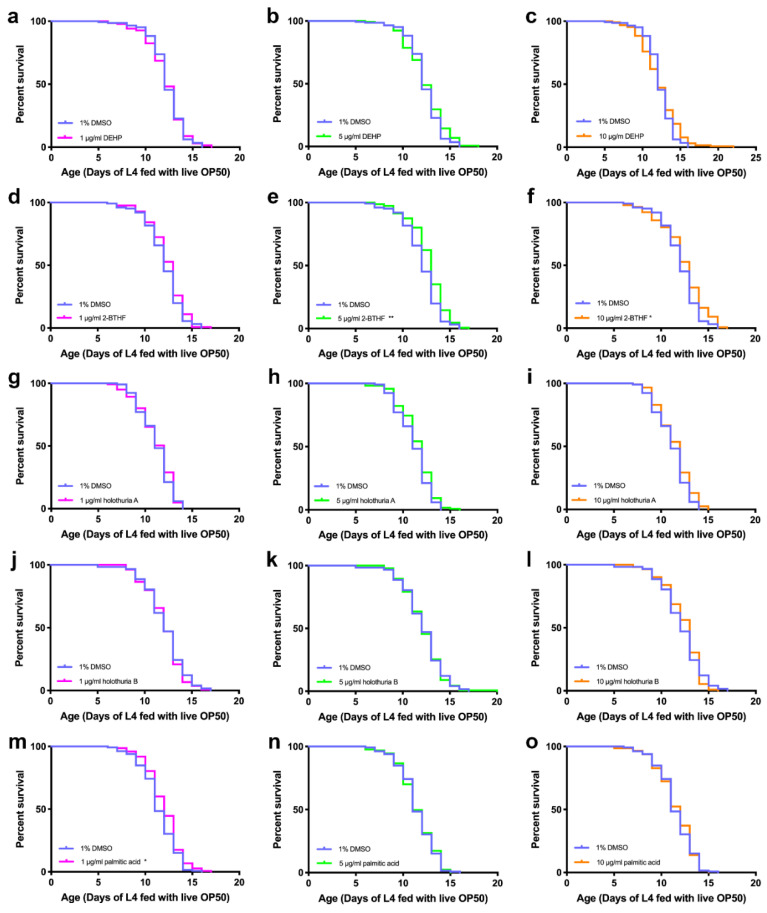
Effects of the compounds derived from *H. scabra* extracts on the lifespan of wildtype N2 worms. The survival curves showed the lifespans of worms treated with 1 μg/mL DEHP (**a**), 5 μg/mL DEHP (**b**), 10 μg/mL DEHP (**c**), 1 μg/mL 2-BTHF (**d**), 5 μg/mL 2-BTHF (**e**), 10 μg/mL 2-BTHF (**f**), 1 μg/mL holothuria A (**g**), 5 μg/mL holothuria A (**h**), 10 μg/mL holothuria A (**i**), 1 μg/mL holothuria B (**j**), 5 μg/mL holothuria B (**k**), 10 μg/mL holothuria B (**l**), 1 μg/mL palmitic acid (**m**), 5 μg/mL palmitic acid (**n**), and 10 μg/mL palmitic acid (**o**) compared to the control. The survival curves were analyzed by using the log-rank (Mantel-Cox) test for *p* values, * *p* < 0.05 and ** *p* < 0.01.

**Figure 2 pharmaceuticals-15-01374-f002:**
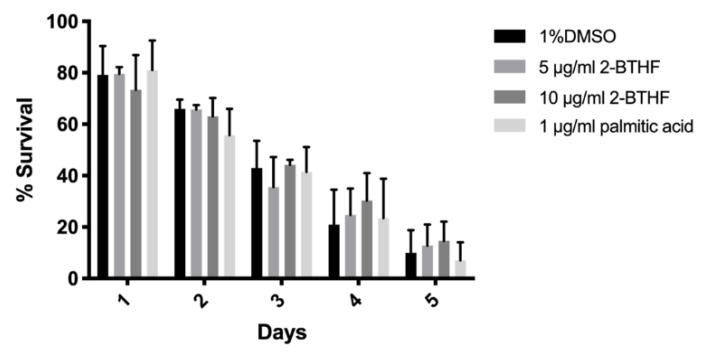
Effects of *H. scabra* compounds on thermal stress resistance in wildtype N2 worms. Thermal stress result was analyzed by two-way ANOVA following Bonferroni’s method.

**Figure 3 pharmaceuticals-15-01374-f003:**
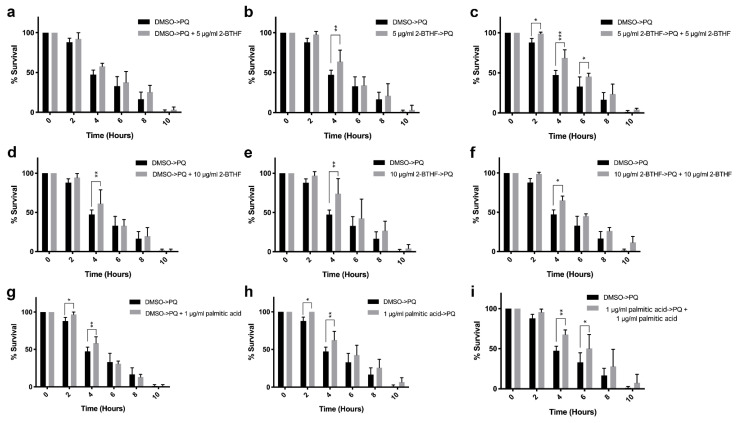
Effects of purified *H. scabra* compounds on wild type N2 worms’ resistance against oxidative stress: the percentages of survival of worms treated with 5 μg/mL 2-BTHF and PQ in co-treatment (**a**), pre-treatment (**b**), and all-treatment (**c**) conditions; the percentages of survival of worms treated with 10 μg/mL 2-BTHF and PQ in co-treatment (**d**), pre-treatment (**e**), and all-treatment (**f**) conditions; the percentages of survival of worms treated with 1 μg/mL palmitic acid and PQ in co-treatment (**g**), pre-treatment (**h**), and all-treatment (**i**) conditions; the percentages of survival of worms treated with 0.1% DMSO and PQ were used as the control. The results of oxidative stress were analyzed by two-way ANOVA following Bonferroni’s method. * *p* < 0.05, ** *p* < 0.01, and *** *p* < 0.001.

**Figure 4 pharmaceuticals-15-01374-f004:**
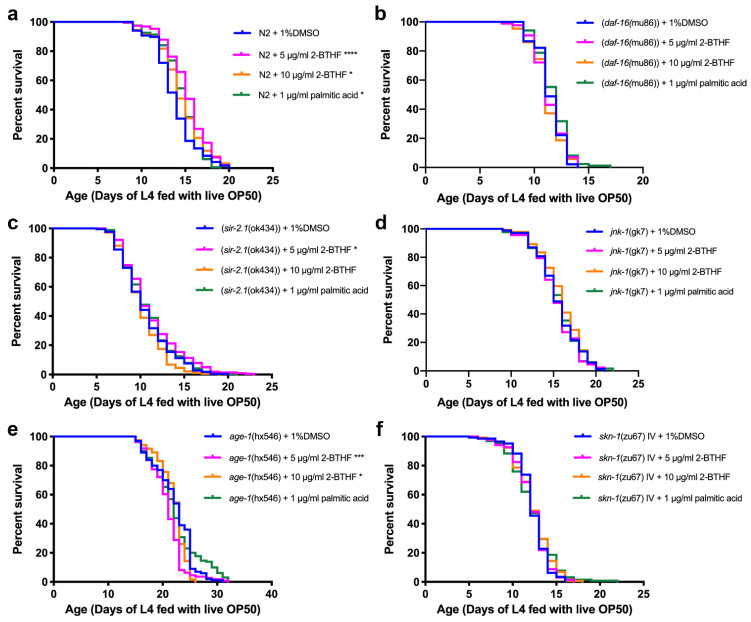
Effects of purified *H. scabra* compounds on the lifespans of wild type and mutant worms. The survival curves of N2 (**a**), CF1038 (*daf-16*(mu86)) (**b**), VC199 (*sir-2.1*(ok434)) (**c**), VC8 (*jnk-1*(gk7)) (**d**), TJ1052 (*age-1*(hx546)) (**e**), and EU1 (*skn-1*(zu67) IV) (**f**) treated with 1% DMSO, 5 μg/mL 2-BTHF, 10 μg/mL 2-BTHF, and 1 μg/mL palmitic acid were shown. The survival curves were analyzed by using log-rank (Mantel-Cox) test for *p* value, * *p* < 0.05, *** *p* < 0.001, and **** *p* < 0.0001.

**Figure 5 pharmaceuticals-15-01374-f005:**
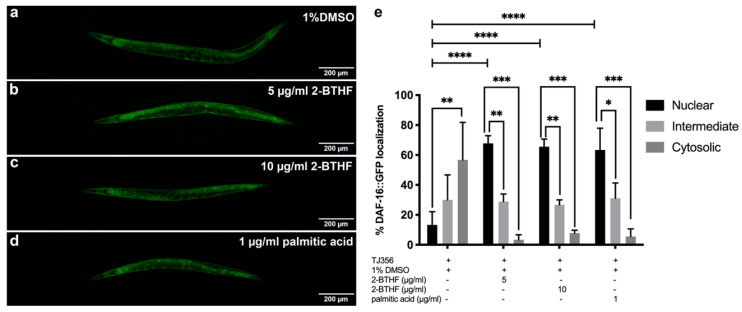
Effects of purified *H. scabra* compounds on DAF-16 nuclear translocation in transgenic TJ356 strain. DAF-16::GFP localizations in the worms treated with 1% DMSO (control) (**a**), 5 μg/mL 2-BTHF (**b**), 10 μg/mL 2-BTHF (**c**), and 1 μg/mL palmitic acid (**d**) are shown. The graph shows percentages of DAF-16::GFP localization of TJ356 worms in nucleus, cytosol, or both (intermediate) (**e**). The result of nuclear translocation was analyzed by two-way ANOVA following Bonferroni’s method, * *p* < 0.05, ** *p* < 0.01, *** *p* < 0.001, **** *p* < 0.0001.

**Figure 6 pharmaceuticals-15-01374-f006:**
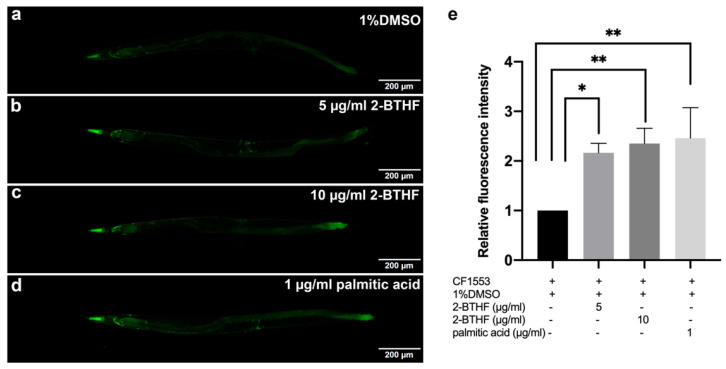
Effects of purified *H. scabra* compounds on SOD-3 expression in transgenic CF1553 strain. GFP expression in the worms treated with 1% DMSO (control) (**a**), 5 μg/mL 2-BTHF (**b**), 10 μg/mL 2-BTHF (**c**), and 1 μg/mL palmitic acid (**d**) extracts are shown. The relative fluorescence intensity of *sod-3p::*GFP was quantified and shown (**e**). The results of SOD-3 GFP expression were analyzed by one-way ANOVA following Bonferroni’s method, * *p* < 0.05 and ** *p* < 0.01.

**Figure 7 pharmaceuticals-15-01374-f007:**
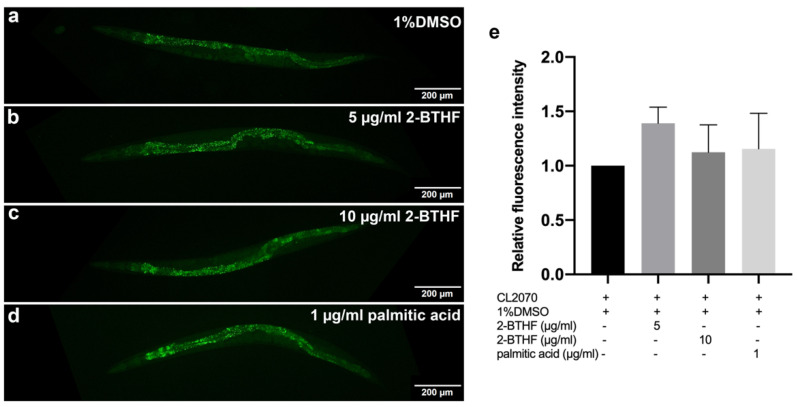
Effects of purified *H. scabra* compounds on HSP-16.2 expression in transgenic CL2070 strain. GFP expression of the worms treated with 1% DMSO (control) (**a**), 5 μg/mL 2-BTHF (**b**), 10 μg/mL 2-BTHF (**c**), and 1 μg/mL palmitic acid (**d**) are shown. The relative fluorescence intensity of *hsp-16.2::*GFP in the worm head region was quantified and shown (**e**). The results of HSP-16.2 GFP expression were analyzed by one-way ANOVA following Bonferroni’s method.

**Figure 8 pharmaceuticals-15-01374-f008:**
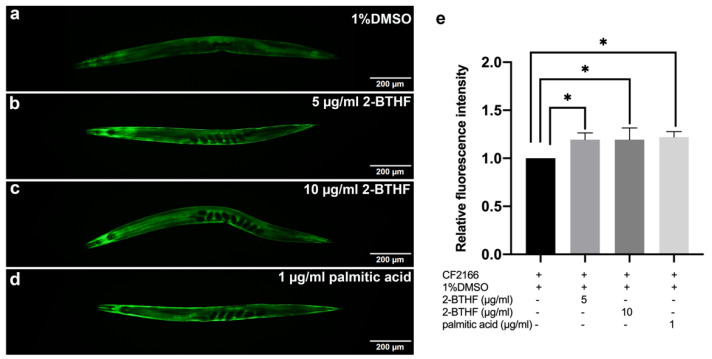
Effects of purified *H. scabra* compounds on GST-4 expression in transgenic CL2166 strain. GFP expression of the worms treated with 1% DMSO (control) (**a**), 5 μg/mL 2-BTHF (**b**), 10 μg/mL 2-BTHF (**c**), and 1 μg/mL palmitic acid (**d**) is shown. The relative fluorescence intensity of *gst-4*p::GFP was quantified and shown (**e**). The results of *gst-*4 GFP expression were analyzed by one-way ANOVA following Bonferroni’s method, * *p* < 0.05.

**Figure 9 pharmaceuticals-15-01374-f009:**
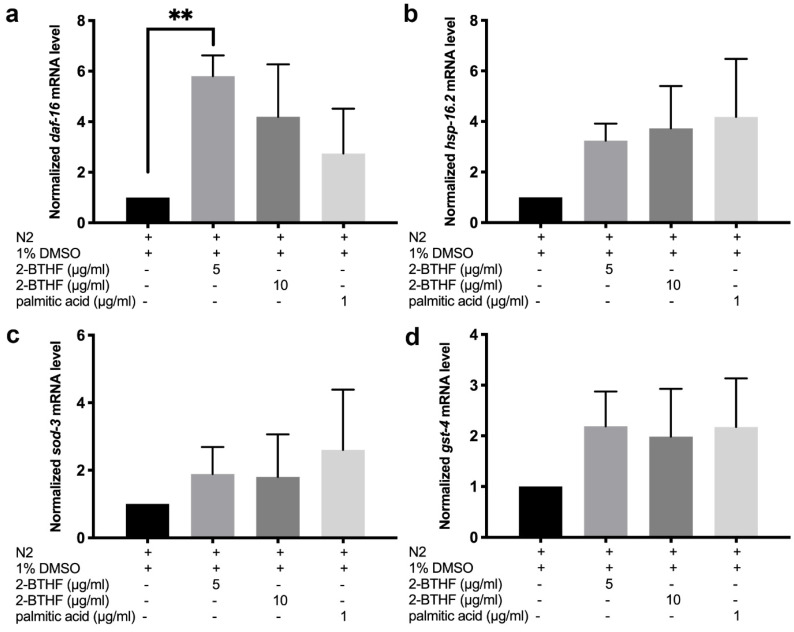
The mRNA levels of *daf-16*, *hsp-16.2*, *sod-3*, and *gst-4* in N2 worms treated with 1% DMSO, 5 μg/mL 2-BTHF, 10 μg/mL 2-BTHF, and 1 μg/mL palmitic acid of *H. scabra* were quantified using qRT-PCR. The normalized mRNA levels of *daf-16* (**a**), *hsp-16.2* (**b**), *sod-3* (**c**), and *gst-4* (**d**) are shown. The results of gene expression were analyzed by one-way ANOVA following Bonferroni’s method, ** *p* < 0.01.

**Table 1 pharmaceuticals-15-01374-t001:** Effects of purified *H. scabra* compounds on mean lifespans of wildtype N2 worms at 25 °C.

Compounds	Lifespan (Mean ± SD)	(Number of Worms, Censored)	% Increase Lifespan	*p* Value (Log-Rank Test)
1% DMSO	12.28 ± 1.76	(145, 8)	-	-
1 μg/mL DEHP	12.18 ± 1.97	(137, 2)	−0.81	0.9737
5 μg/mL DEHP	12.36 ± 2.10	(145, 4)	0.65	0.2603
10 μg/mL DEHP	12.29 ± 2.49	(129, 2)	0.08	0.2695
1% DMSO	12.04 ± 1.91	(126, 2)	-	-
1 μg/mL 2-BTHF	12.35 ± 1.86	(127, 9)	2.57	0.1694
5 μg/mL 2-BTHF	12.72 ± 1.90	(151, 2)	5.65	0.0012 **
10 μg/mL 2-BTHF	12.37 ± 2.36	(142, 3)	2.74	0.0192 *
1% DMSO	11.10 ± 1.70	(118, 2)	-	-
1 μg/mL holothuria A	11.13 ± 1.88	(121, 2)	0.27	0.5654
5 μg/mL holothuria A	11.45 ± 1.79	(118, 1)	3.15	0.0895
10 μg/mL holothuria A	11.44 ± 1.77	(123, 3)	3.06	0.0703
1% DMSO	12.12 ± 2.14	(123, 0)	-	-
1 μg/mL holothuria B	12.07 ± 1.91	(134, 0)	−0.41	0.5685
5 μg/mL holothuria B	12.17 ± 2.00	(134, 2)	0.41	0.8980
10 μg/mL holothuria B	12.30 ± 1.85	(112, 1)	1.49	0.8498
1% DMSO	11.45 ± 1.87	(132, 4)	-	-
1 μg/mL palmitic acid	11.99 ± 1.86	(148, 2)	4.72	0.0207 *
5 μg/mL palmitic acid	11.46 ± 1.92	(127, 0)	0.09	0.8550
10 μg/mL palmitic acid	11.48 ± 1.96	(145, 0)	0.26	0.8037

* and ** Indicates significant increase in mean lifespan between the treated group and the control group at *p* < 0.05 and *p* < 0.01.

**Table 2 pharmaceuticals-15-01374-t002:** Effects of purified *H. scabra* compounds on mean lifespans of wild type N2 and mutant worms at 22 °C.

Strains of Worm	Compounds	Lifespan (Mean ± SD)	(Number of Worms, Censored)	% Increase Lifespan	*p* Value (Log-Rank Test)
N2	1% DMSO	13.79 ± 2.40	(118, 0)	-	-
	5 μg/mL 2-BTHF	15.20 ± 2.34	(190, 0)	10.22	<0.0001 ****
	10 μg/mL 2-BTHF	14.50 ± 2.60	(126, 0)	5.15	0.0174 *
	1 μg/mL palmitic acid	14.49 ± 2.11	(163, 0)	5.08	0.0422 *
CF1038 (*daf-16*(mu86))	1% DMSO	11.42 ± 1.30	(90, 0)	-	-
	5 μg/mL 2-BTHF	11.33 ± 1.47	(86, 0)	−0.79	0.9129
	10 μg/mL 2-BTHF	11.16 ± 1.49	(86, 0)	−2.28	0.3820
	1 μg/mL palmitic acid	11.73 ± 1.48	(85, 0)	2.71	0.1169
VC199 (*sir-2.1*(ok434))	1% DMSO	10.51 ± 2.86	(215, 0)	-	-
	5 μg/mL 2-BTHF	11.08 ± 3.26	(203, 0)	5.42	0.0416 *
	10 μg/mL 2-BTHF	10.13 ± 2.36	(178, 0)	−3.62	0.0799
	1 μg/mL palmitic acid	10.81 ± 2.86	(266, 1)	2.85	0.2841
VC8 (*jnk-1*(gk7))	1% DMSO	15.51 ± 2.46	(176, 2)	-	-
	5 μg/mL 2-BTHF	15.32 ± 2.44	(92, 0)	−1.23	0.5789
	10 μg/mL 2-BTHF	15.91 ± 2.43	(139, 1)	2.58	0.1965
	1 μg/mL palmitic acid	15.52 ± 2.49	(133, 0)	0.06	0.8594
TJ1052 (*age-1*(hx546))	1% DMSO	22.16 ± 3.60	(100, 0)	-	-
	5 μg/mL 2-BTHF	20.87 ± 3.06	(111, 0)	−5.82	0.0002 ***
	10 μg/mL 2-BTHF	21.96 ± 2.57	(119, 0)	−0.90	0.0136 *
	1 μg/mL palmitic acid	22.29 ± 4.39	(130, 0)	0.59	0.6533
EU1 (*skn-1*(zu67) IV)	1% DMSO	12.28 ± 1.76	(145, 0)	-	-
	5 μg/mL 2-BTHF	12.18 ± 1.97	(137, 0)	−0.81	0.9737
	10 μg/mL 2-BTHF	12.36 ± 2.10	(145, 0)	0.65	0.2603
	1 μg/mL palmitic acid	12.29 ± 2.49	(129, 0)	0.08	0.2695

*, ***, **** Indicates significant difference of mean lifespan between the treated group and the control group at *p* < 0.05, *p* < 0.001, and *p* < 0.0001, respectively.

**Table 3 pharmaceuticals-15-01374-t003:** The primers used for qRT-PCR experiment.

Genes	Forward Primers	Reverse Primers
*daf-16*	5′-CCAGACGGAAGGCTTAAACT-3′	5′-ATTCGCATGAAACGAGAATG-3′
*hsp-16.2*	5′-GTCACTTTACCACTATTTCCGT-3′	5′-CAATCTCAGAAGACTCAGATGG-3′
*sod-3*	5′-CCAACCAGCGCTGAAATTCAATGG-3′	5′-GGAACCGAAGTCGCGCTTAATAGT-3′
*gst-4*	5′- CCCATTTTACAAGTCGATGG-3′	5′-CTTCCTCTGCAGTTTTTCCA-3′
*act-1*	5′-ATCGTCACCACCAGCTTTCT-3′	5′-CACACCCGCAAATGAGTGAA-3′

## Data Availability

Data is contained within the article.

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
