# Peer review of "2-Butoxytetrahydrofuran and Palmitic Acid from *Holothuria scabra* Enhance *C. elegans* Lifespan and Healthspan via DAF-16/FOXO and SKN-1/NRF2 Signaling Pathways"

_pharmaceuticals, 2022, doi:10.3390/ph15111374_

Round 1

Reviewer 1 Report

Account

To the article of the authors   P. Jattujan, S. Srisirirung, W. Watcharaporn ET AL. -@Butoxytetrahydrofuran and palmitic acid from Holothuria scabra enhance C. elegans lifespan and healthspan via DAF 16/FOXO and SKN-1/NRF2 signaling pathways 4

The present article is dedicated to yhe Extracts from a sea cucumber, Holothuria scabra, have been shown to exhibit various pharmacological properties including anti-oxidation, anti-aging, anti-cancer, and anti-neurodegeneration. Furthermore, certain purified compounds from H. scabra displayed neuroprotective effectsagainst Parkinson’s and Alzheimer’s diseases. Therefore, in the present study we further examined the anti-aging activity of purified H. scabra compounds in Caenorhabditis elegans model. Five com pounds were isolated from ethyl acetate and butanol fractions of the body wall of H. scabra and characterized as holothuria A, holothuria B, palmitic acid, bis (2-ethylhexyl) phthalate (DEHP) and-butoxytetrahydrofuran (2-BTHF).

Thus, the presented review article is of undoubted interest and therefore it can be published in the Journal “Pharmaceuticals”

Author Response

We thank the reviewer for your kind words and encouragements to our manuscript. We wish that our manuscript provides the high impact to the society and support the aims and scopes of the Journal “Pharmaceuticals”.

Reviewer 2 Report

Jattujan, P. et al demonstrate the effect of 2-Butoxytetrahydrofuran and palmitic acid from Holothuria scabra on enhance the lifespan of C. elegans via DAF-16/FOXO and SKN-1/Nrf2 signaling pathways. The manuscript depicts very interesting findings, but several issues should be addressed.

1. Figure 7. statistical significance data is missing 

2. Figure 9. statistical significance data is missing

Author Response

We thank the reviewer for the comments to improve our manuscript. The results in Figure 7 showed no significant difference, thus we mentioned the p values in line 277 in the revised manuscript. In Figure 9a, only 5 μg/ml 2-BTHF significantly increased daf-16 mRNA expression level when compared to control. However, hsp-16.2, sod-3, and gst-4 mRNA expressions in 2-BTHF and palmitic acid treated-groups were not significant increase (Figure 9b-9c). The specific p values of these data were presented in the corresponding sentences in lines 305-319.

Reviewer 3 Report

The authors performed a study entitled "2-Butoxytetrahydrofuran and palmitic acid from Holothuria scabra enhance C. elegans lifespan and healthspan via DAF-16/FOXO and SKN-1/NRF2 signaling pathways "

In my opinion, the manuscript is well written, the introduction is satisfactory, the results are concise and the discussion has been carried out, in addition, the materials and methods have good supporting references.

I only recommend that the authors improve the conclusion, do not repeat data already presented and discussed in previous sections, and show the advances that their results are bringing to the literature.

Author Response

We thank the reviewer for your kind words and suggestions. We have edited the conclusion as suggested and appeared in lines 591-597 in the revised manuscript.

Reviewer 4 Report

The article gives insights into the purified compounds from sea cucumber to have an anti-aging effect by modulating longevity pathways. Specifically, they demonstrated ways in the nematode model. The article can be accepted for publication.

Author Response

We thank the reviewer for your kind words and encouragements to our manuscript. We wish that our manuscript provides the high impact to enhance the anti-aging research for the society.

Reviewer 5 Report

This is an interesting paper, worth publishing, but in our view the following aspects should be solved before publication:

Line 24: I am afraid that these substances names are written wrongly. Holothuria is the genus of the organism, whereas its toxic compounds are known as “holothurin A” and “holothurin B”. The same for lines 90, 443 and other places where the names of the two compounds occur.

Line 81: mTOR stands for “mammalian target of rapamycin (mTOR)” not “mechanistic target of rapamycin (mTOR)”.

Lines 189-197: it is rather curious that only for the 5 μg/ml 2-BTHF a statistical significance was observed. The authors should try explaining why in the case of the higher concentration (10 μg/ml 2-BTHF) sir-2 was involved, but not in the lower concentration.  

Line 517: the statistical inferential methods used to process the findings are even more important than the statistical software. Therefore, the authors should shortly describe the statistical approach used in their analysis.

Line 550: “RNA miniprep kit with column”. Please clarify the provider of the kit.

Line 554: “specific qPCR primers”. Please clarify what primers were used (reference Table 3 here, if those were used) and clarify how they were sourced.

Lines 274-275: “with no significant difference when compared to the control 1% DMSO group (1.00 ± 0.00)  (p > 0.05)”. Please provide the estimated p value instead of a non-informative “p > 0.05” (0.051 is very different from 0.51, although both are > 0.05). The same for the non-significant values in lines 295-311.

Lines 396-399: this paragraph brushes away the discussions regarding the risks of saturated fatty acids on health and longevity, as well as previous research indicating that saturated fatty acids (including palmitic acid) may result in shorter life-spans (e.g. in rodents and Drosophila, see. Driver, C. J., & Cosopodiotis, G. (1979). The effect of dietary fat on longevity of Drosophila melanogaster. Experimental gerontology14(3), 95-100; see, also Ueda, Y., Wang, M. F., Irei, A. V., Sarukura, N., Sakai, T., & Hsu, T. F. (2011). Effect of dietary lipids on longevity and memory in the SAMP8 mice. Journal of nutritional science and vitaminology57(1), 36-41 and Jové, M., Naudí, A., Aledo, J. C., Cabré, R., Ayala, V., Portero-Otin, M., ... & Pamplona, R. (2013). Plasma long-chain free fatty acids predict mammalian longevity. Scientific reports, 3(1), 1-8.). In our view it is important to discuss previous research and try making sense of it (maybe findings relevant for Caenorhabditis are not necessarily relevant for humans and other animal species?).

Author Response

Please see our response to reviewer in the attachment.

Round 2

Reviewer 5 Report

The paper can now be published in its present form.